# The Impact of Maternal and Piglet Low Protein Diet and Their Interaction on the Porcine Liver Transcriptome around the Time of Weaning

**DOI:** 10.3390/vetsci8100233

**Published:** 2021-10-14

**Authors:** Kikianne Kroeske, Ester Arévalo Sureda, Julie Uerlings, Dieter Deforce, Filip Van Nieuwerburgh, Marc Heyndrickx, Sam Millet, Nadia Everaert, Martine Schroyen

**Affiliations:** 1Precision Livestock and Nutrition Unit, Gembloux Agro-Bio Tech, TERRA Teaching and Research Centre, University of Liège, 5030 Gembloux, Belgium; kikianne.kroeske@gmail.com (K.K.); Ester.ArevaloSureda@uliege.be (E.A.S.); julie.uerlings@gmail.com (J.U.); Nadia.everaert@uliege.be (N.E.); 2Flanders Research Institute for Agriculture, Fisheries and Food (ILVO), 9090 Melle, Belgium; marc.heyndrickx@ilvo.vlaanderen.be (M.H.); sam.millet@ilvo.vlaanderen.be (S.M.); 3Laboratory of Pharmaceutical Biotechnology, Faculty of Pharmaceutical Sciences, Ghent University, 9000 Ghent, Belgium; dieter.deforce@ugent.be (D.D.); Filip.VanNieuwerburgh@UGent.be (F.V.N.); 4Department of Pathology, Bacteriology and Poultry Diseases, Ghent University, 9820 Merelbeke, Belgium; 5Department of Nutrition, Genetics and Ethology, Ghent University, 9820 Merelbeke, Belgium

**Keywords:** pig nutrition, gene expression, maternal effect, late gestation diet, nursery diet, protein

## Abstract

Maternal diet during early gestation affects offspring phenotype, but it is unclear whether maternal diet during late gestation influences piglet metabolism. We evaluated the impact of two dietary protein levels in sow late gestation diet and piglet nursery diet on piglet metabolism. Diets met or exceeded the crude protein and amino acid requirements. Sows received either 12% (Lower, L) or 17% (Higher, H) crude protein (CP) during the last five weeks of gestation, and piglets received 16.5% (L) or 21% (H) CP from weaning at age 3.5 weeks. This resulted in a 2 × 2 factorial design with four sow/piglet diet treatment groups: HH and LL (match), HL and LH (mismatch). Piglet hepatic tissues were sampled and differentially expressed genes (DEGs) were determined by RNA sequencing. At age 4.5 weeks, 25 genes were downregulated and 22 genes were upregulated in the mismatch compared to match groups. Several genes involved in catabolic pathways were upregulated in the mismatch compared to match groups, as were genes involved in lipid metabolism and inflammation. The results show a distinct interaction effect between maternal and nursery diets, implying that sow late gestation diet could be used to optimize piglet metabolism.

## 1. Introduction

The dietary composition of late gestation diets has been studied in several species because of its impact on offspring [1,2,3]. Protein levels in the maternal diet influence the offspring’s glucocorticoid sensitivity as well as lipid, triglyceride, and cholesterol metabolism [4,5]. This relates to the concept of metabolic programming, which assumes that the ability of the offspring to digest feed in a certain way is (partly) determined by maternal diet [3,6]. This metabolic programming can be reflected in gene expression differences and could cause piglets to be more or less feed- or growth-efficient. Studies in pigs and other species have offered insights on maternal dietary influences on offspring performance [7,8,9]. The perinatal period was shown to be critical, including for muscle development and fat metabolism in piglets [8,10]. It is known that maternal protein diet influences offspring metabolism from conception onwards, as shown in studies with deficient or excessive dietary CP [11]. At the same time, it has become more common to produce piglet feed with low dietary protein while maintaining adequate amino acid levels. Adequate protein and amino acid levels are necessary for lean meat development, but lower protein diets are preferred to decrease urea excretion [12]. Moreover, high protein diets have been shown to increase post-weaning diarrhea [1,13]. However, the influence of high or low crude protein diets with adequate amino acid levels during late gestation in sows on the metabolic processes of piglets is not yet fully elucidated. 

Deficient levels of dietary crude protein alter the liver transcriptome of pigs [7,11]. The liver is a strong indicator of metabolic processes because the liver has an essential role in detoxifying, metabolizing, and distributing nutrients. Nutrients are transported from the gut through the hepatic vein into the liver and the liver regulates nutrient partitioning [14]. In the liver, amino acids are not only converted into proteins but can also be metabolized into fatty acids and glucose and stored in the body as lipids or glycogen, as the body cannot store amino acids and excess nitrogen is disposed of as urea [14]. Differentially expressed genes due to different protein diets have been shown to alter pathways in glucose metabolism [15], fat metabolism, and endocrine regulation [16]. 

In an earlier publication, we showed that the sow diet in late gestation affects their offspring until slaughter. We focused on protein diets that met all nutrient requirements; all diets were thus formulated according to the recommendations of the NRC [17] and CVB [18]. We observed that reducing protein levels in the maternal late gestation diet and increasing protein levels in the nursery piglet diet (LH) led to a higher, and thus less desirable, fecal consistency score and worse feed efficiency. In addition, serum urea concentrations tended to be lower for the piglets that received the same treatment as their mother (“match” groups, both on a high or a low protein diet) at 23 weeks of age [19]. The object of the current study was to gain more insight into how maternal late gestation diet influences the piglets’ metabolic programming in the first week after weaning and how this interacts with their own diet during the nursery phase. We hypothesized that the maternal protein level during late gestation affects (or programs) the transcriptome of piglets. 

## 2. Materials and Methods

### 2.1. Experimental Design and Animals

The experimental design is described in detail in a previous report [19]. In short, all sows were inseminated ad random with semen from either one of two Piétrain boars and were divided over two batches with a three-week interval. The pregnant sows (*n* = 35) were divided into two dietary treatments: low (L) 12% CP or high (H) 17% CP starting at 5 weeks before the expected farrowing date. All feeds had similar concentrations of essential amino acids and were iso-energetic (6.5 standardized ileal digestible lysine (SIDLYS) g/kg, 9.1 MJ/kg net energy). After farrowing, all sows received the same lactation feed. Piglets did not receive creep feed during the lactation period. Each nursery pen in the experiment consisted of the 6 sibling piglets closest to the average body weight per pen (3 gilts and 3 barrows, n_total_ = 209, 35 pens). Due to a shortage of female piglets in one litter, one pen contained only 2 gilts and 3 barrows. From weaning at 3.5 weeks to 9 weeks of age, each pen was assigned and evenly distributed (random stratified method) to a low (L) 16% CP or a high (H) 21% CP diet with similar levels of essential amino acids and energy levels (10.5 g SID LYS/kg, 9.8 MJ/kg) [18]. Each litter was already divided according to sow dietary treatment (*n* = 17 L sow and *n* = 18 H sow) and were subsequently randomly assigning piglets to dietary treatment. This resulted in a stratified randomized 2 × 2 factorial design with four groups: high protein level in sow and piglet diet (HH), high protein level in sow diet and low protein level in piglet diet (HL), low protein level in sow diet and high protein level in piglet diet (LH), low protein level in sow and piglet diet (LL). At 3.5 weeks of age (weaning age), 24 piglets (all 24 animals originate from different sows, *n* = 12 for low (L) sow diet, *n* = 12 for high (H) sow diet, equally divided by sex) and at 4.5 weeks of age, 32 piglets (*n* = 8 for each treatment HH, HL, LH and LL, all-male, all from different sows) were anesthetized using a Zoletil 100^®^ (Virbac, Louvain la Neuve, Belgium) and Xyl-M^®^ 2% (VMD, Arendonk, Belgium) intramuscular injection and euthanized using an intracardiac injection of Release^®^ 300 mg/mL (ECUPHAR NV/SA, Oostkamp, Belgium). Blood was collected intracardially from piglets that were to be euthanized. After sampling, blood was kept on ice for approximately 1–4 h until serum was obtained by centrifuging at 1500× *g* for 10 min at 4 °C, and subsequently serum was frozen at −80 °C until further analysis. Liver samples were taken within 20 min post mortem, dissected from the middle of the right lobe, snap-frozen in liquid nitrogen, and stored at −80 °C until further analysis.

### 2.2. RNA Extraction and Sequencing

Liver samples were ground into a fine powder in liquid nitrogen, and total RNA was extracted using the Promega ReliaPrep™ RNA Tissue Miniprep system according to the manufacturer’s protocol, including a DNase treatment (Promega Corporation, Madison, WI, USA). The concentration and quality of the extracted total RNA were checked using the Quant-it ribogreen RNA assay (Life Technologies, Grand Island, NY, USA) and the LabChip GX RNA kit (Perkin Elmer, Waltham, MA, USA). Subsequently, Illumina sequencing libraries were prepared from 200 ng of RNA of each sample, using the QuantSeq 3’ mRNA-Seq FWD Library Prep Kit (Lexogen, Vienna, Austria) according to the manufacturer’s protocol using 15 enrichment PCR cycles. Samples were not pooled prior to library preparation. Libraries were quantified by qPCR, according to Illumina’s ‘Sequencing Library qPCR Quantification protocol guide’, version February 2011. The LabChip GX High sensitivity DNA kit (Perkin Elmer, Waltham, MA, USA) was used to control the size distribution and quality of the library. Libraries were equimolarly pooled and sequenced on a NextSeq 500, generating 75 base pair single-end reads. The sequencing yielded 6.85 × 10^6^ ± 1.72 × 10^6^ (mean ± standard deviation) raw reads per sample. The mean Phred score across each base position in the reads was >28 for all samples (Appendix A). The raw reads are processed by trimming the adapter sequences and polyA tail sequences using Trim Galore [20]. The length distribution of the remaining reads can be found in Appendix A. The trimmed reads were filtered to remove rRNA reads using Bowtie2 [21] and a custom-built reference database with all pig ribosomal RNA sequences found on SILVA [22,23]. The 6.67 × 10^6^ ± 1.67 × 10^6^ remaining reads were mapped against the Sus scrofa v11.1 reference genome [24] from the Ensembl genome database using the STAR mapper [25]. On average 78.51% of the reads mapped uniquely to the reference genome, 87.12% of the reads mapped uniquely or to a limited number of loci in the reference genome.

### 2.3. RNA-Seq Differential Expression Analysis

The R-package DESeq2 (1.30.1) was employed to identify differentially expressed genes (DEGs) [26]. DESeq2 applied standard normalization. The data was normalized by the DEseq2 function ‘DESeq’ with a normalization factor s_ij_, i.e., μ_ij_ = s_ij_q_ij_ [26]. Genes were considered as DE when the difference between treatments was significant after adjusting for multiple testing using Benjamini–Hochberg (FDR < 0.05). However, for pathway analyses, genes differential expressed at an FDR < 0.10 were considered. At 3.5 weeks of age, the model was as follows (1).
DESeq2 model design 3.5 weeks of age =~ Sex + Batch + Boar + Diet_sow_(1)

‘Batch’ was included as a fixed factor in the model because initial analysis showed a strong separation by batch. ‘Boar’ was included as a fixed factor because two boars were used with insemination and literature shows increasing evidence of paternal influence at conception on DEGs in offspring [27,28]. At 4.5 weeks of age, sow diet and piglet diet were modeled in an interaction-effect design (2).
DESeq2 model design 4.5 weeks of age =~ Batch + Boar + Diet_sow_ + Diet_piglet_ + Diet_sow_:Diet_piglet_(2)

The result of Formula 2 resulted in four treatment groups, the list of genes was exported, visualized and the DEGs were categorized as ‘match’ or ‘mismatch’. DEGs with a count lower than 10 in more than 90% of the samples were filtered out before the DEG analysis. After generating the interaction-affected DEGs, we also examined the main effect for both sow and piglet diets using the same statistical model. A ’main effect’ in this paper is defined as the effect by only one of either diet, i.e., sow or piglet diet. The remaining DEGs were exported and pathway analysis was performed using RStudio package g:profiler2 (version 0.2.1) with Sus scrofa as the reference organism [29]. Ensembl IDs were directly imported from STAR mapping, using the g:Profiler2 database. DEGs from the interaction effect had an FDR < 0.05. For the pathway analyses, genes were used with an FDR < 0.10. This high FDR allowed us to obtain more genes for a proper, more meaningful pathway analysis. Pathways were only included when 5 or more DEGs were representing the pathway. DEGs of main effects are expressed as L vs. H, meaning that the relative difference is found in the low protein diet compared to the high protein diet for both sow and piglet diets unless stated otherwise. The normalized counts were used for further calculations.

### 2.4. Primer Design and qPCR Validation

For the validation process, RNA was converted into cDNA by reverse transcription of 50 ng/µL total RNA using GoScript^TM^ Reverse Transcription Mix with Random Primers (Promega Corporation, Madison, WI, USA). For four genes of interest, gene-specific primers were designed using Primer3 (Appendix A). ACTB and YWHAZ were employed as housekeeping genes as they were found to be the most stable genes according to the RNA-seq counts. Quantitative PCR (qPCR) was performed using the Takara TB Green^®^ Premix EX TaqTM II (200 Rxns) (Takara Bio Europe AB, Gothenburg, Sweden) and a ThermoFisher QuantStudio™ 3 System (Thermo Fisher, Waltham, MA, USA). The qPCR reactions were carried out in a final volume of 20 μL consisting of 2 μL cDNA, 10 μL of SYBR, 0.4 μL ROX, 1 μL (0.5 μM) of both forward and reverse primer, and 5.6 μL molecular grade water. All runs were performed with a default qPCR protocol, starting with initial denaturation (30 s, 95 °C), followed by amplification for 40 cycles (95 °C for 5 s, 60 °C for 30 s, and 72 °C for 45 s). Primer specificity was verified through melting curves. All primers were tested: efficiency was between 90% and 113%. The normalized values were used to calculate the relative gene expression levels using the 2^-ΔΔct^ method [30]. Results were statistically analyzed using numSummary (RcmdrMisc, R), and mixed models (lme4, R) with the interaction effect of piglet and sow diet (multiplied within the model) as fixed effects and Batch as a random effect with a significance threshold of *p* < 0.05. Spearman correlations were acquired by comparing the normalized RNA-seq counts with the relative expression levels obtained by qPCR.

### 2.5. Serum Amyloid A Immunoassay 

Serum amyloid A (SAA) levels were analyzed using a commercial sandwich enzyme-linked immunosorbent assay (Solid Phase ELISA, Tridelta Development Ltd., Maynooth, Co. Kildare, Ireland). Samples were tested in triplicate in dilutions ranging from 1:500 to 1:10 to ensure measurement of low concentrations. Dilutions were determined based on signal and repeated if there was no or an inconsistent signal. One sample from the HL group was hemolyzed and was therefore excluded from statistical analysis. The standard curve consisted of a seven-step serial dilution ranging from 1000 to 31.25 ng/mL, and samples below the detection limit were counted as zero. Results were statistically analyzed using numSummary (RcmdrMisc, R), and mixed models (lme4, R) with the interaction effect of piglet and sow diet (multiplied within the model) as fixed effects and Batch as a random effect with a significance threshold of *p* < 0.05. Quantitative correlations were calculated using Spearman’s method. Log-normal distribution was possible by adding 0.01 to all values (solving the problem of many zero values) and log-scaling the function within a glm model. Values that were originally zero were still counted as zero. All values were calculated, corrected for the dilution step, expressed in μg/mL, and shown on a log scale. 3. [30]. 

## 3. Results

### 3.1. Differentially Expressed Genes

In this experiment, 17,901 transcripts were successfully mapped to the Sus scrofa reference genome (v11.1). At 3.5 weeks of age, 3 genes were differentially expressed due to the sow diet: 2 were downregulated and 1 was upregulated in the H sow diet group as compared to the L sow diet group. No genes were of interest. The small number of DEGs made it impossible to perform pathway analysis (Appendix A).

At 4.5 weeks of age, 47 genes were differentially expressed in the mismatch (HL and LH) groups compared to the match (HH and LL) groups. This was divided into 25 downregulated genes in the mismatch compared to the match groups while 22 genes were upregulated (Table 1). Table 1 includes a column indicating whether genes were upregulated or downregulated in the mismatch compared to the match groups. However, to provide enough genes for the pathway analyses, we examined all 313 DE genes with an FDR < 0.10. Of these, 164 of these were downregulated and 149 were upregulated in the mismatch groups (Table 2). Some interesting genes that were upregulated in mismatch groups were Forkhead box A1 (FOXA1), Prospero homeobox 1 (PROX1), Serum amyloid A (SAA2), hepatocyte nuclear factor 4 alpha (HNF4α), and 25-hydroxycholesterol 7-alpha-hydroxylase-like (CYP7B1). For the main effect of the sow diet at 4.5 weeks of age, no DEGs were found and for the main effect of piglet diet only 3 DEGs were found, 2 of which are upregulated in H piglet (Appendix A). In addition, when taken the FDR cut-off at 0.10, 5 differential pathways were found for piglet diet main effect, 3 of which were upregulated in H piglets (Appendix A).

### 3.2. qPCR Results

Four DEGs from the 4.5 weeks of age dataset were selected for qPCR validation due to their potential biological relevance for metabolism: SAA2, FOXA1, PROX1, and the gene encoding for the RING1 and YY1 binding protein (RYBP). The results were analyzed per treatment group at 4.5 weeks of age. All 4 genes showed a significant interaction value (P_s*p_) which corresponds to the RNA-seq analysis (Figure 1). Significant Spearman correlations between the RNA-seq and qPCR analysis were found for all 4 genes: SAA2 (*r* = 0.97, *p* < 0.001), FOXA1 (*r* = 0.80, *p* < 0.001), PROX1 (*r* = 0.61, *p* < 0.001), and RYBP (*r* = 0.65, *p* < 0.001). 

### 3.3. SAA-ELISA 

The SAA-ELISA (*n* = 55 serum samples) resulted in 41 positive values, with 40 of these samples above the limit of quantification. No difference was found at 3.5 weeks of age for sow treatment (Table 3). A tendency to an interaction between sow and piglet diet was observed at 4.5 weeks of age in the ELISA with a higher average value in the HL group compared to the other groups (*P*_s*p_ = 0.07, Table 4). The SAA counts from the RNA seq correlated with the SAA-ELISA results at 3.5 weeks of age with a correlation coefficient *r* = 0.75 (*P* < 0.01), and at 4.5 weeks of age with *r* = 0.81 (*P* < 0.01).

## 4. Discussion

During late gestation, protein requirements of sows increase as compared to early gestation because of the nutritional support needed for fetal and mammary gland development [31]. A previous report shows that piglets with additional protein supplementation from the mother during the fetal phase are likely to be programmed for metabolizing high protein diets and are therefore more likely to more efficiently digest excessive amounts of dietary protein. In the same experiment, the late gestation diet affected offspring until slaughter, with slightly better feed efficiency for piglets with HH and LL diets observed from 3.5 until 15 weeks of age [19]. In the present report, we have described the underlying mechanisms for this crossover effect between high to low or low to high protein diet that would affect a piglet’s metabolism on a transcriptomic level. 

### 4.1. Interaction Effects—Pathway Analyses

The number of DEGs found at 4.5 weeks of age (comparison of mismatch (HL and LH) to match groups (HH and LL)) was large enough to perform pathway analyses when taken at an FDR < 0.10. Eight pathways relating to catabolic processes were found upregulated in the mismatch groups. One was the ’ubiquitin-dependent protein catabolic process’ (GO:0006511, Table 2) which codes for the breakdown of proteins and peptides by hydrolysis of peptide bonds. Other catabolic processing pathways were upregulated in the mismatch groups as well, including ‘modification-dependent protein catabolic process (GO:0019941)’, ‘modification-dependent macromolecule catabolic process (GO:0043632)’, and ‘proteasomal protein catabolic process (GO:0010498)’. It must be stated that these pathways are also part of intracellular degradation processes and might not be involved in the utilization of dietary protein directly. However, the differences shown are in the function of our dietary treatment and must also be mentioned. On the other hand, when we provided the offspring from H- sows with a high protein or amino acid supplementation during the weaning phase, the offspring showed less catabolic metabolism and increased feed efficiency [19]. This suggests that these HH piglets were programmed to metabolize excess amounts of protein more efficiently during an unknown period of their lives. The piglets from H sows subsequently given a low protein diet (HL) had the same starting point at 3.5 weeks of age as HH piglets but received less dietary protein and showed increased protein catabolic rates, which lead to decreased feed efficiency. Increased feed efficiency was also revealed to increase protein metabolism in the liver transcriptome [32]. Piglets from an L sow are not programmed during their fetal development for high protein breakdown and are therefore most likely not having an increased protein breakdown in the liver, thus also resulting in a decreased feed efficiency, which is the ratio between feed intake and body weight gain, the latter influenced by their metabolism. The upregulated pathways for the match groups compared to the mismatch groups were mostly related to general RNA and protein metabolism. Thus, mismatching protein levels in late gestation and weaning diet is more likely to upregulate catabolic pathways. 

One study with a similar setup found an effect of maternal late gestation diet supplemented with uridine [2]. They demonstrated an effect on placental nutrient transportation, largely in response to an alteration of the mTORC1–PPAR signaling pathway, and they observed an overall improvement of the reproductive performance in the sows. Concerning the offspring, they noted an upregulated nutritional metabolism, showing an increase in fatty acid and amino acids metabolism in the liver of neonatal piglets [2]. Although the maternal late-gestation dietary influences were only minor for the study of Gao et al. [2] and in the present study, they are observed also and might need consideration when designing feeding strategies. An important but unanswered question is how long the effects persist over time.

### 4.2. Interaction Effect—Individual Genes

Forkhead box A1 (FOXA1) and Prospero Homeobox 1 (PROX1) are involved in fat metabolism and were both increased in the mismatch groups. FOXA1 reduces lipid accumulation in hepatocytes and PROX1 has been described to modulate lipid homeostasis [33,34]. Depleting PROX1 results in an increased number of triglycerides in the liver [34]. PROX1 and histone deacetylase 3 (HDAC3) are co-recruited by hepatocyte nuclear factor 4α (HNF4α) and HNF4α is upregulated at an FDR of 0.10 in the mismatch groups in the present study. The PROX1-HDAC3 complex controls the expression of other genes regulating lipid homeostasis in the liver [34]. However, we did not find HDAC3 to be differentially expressed. The upregulation of two out of three important gene regulators for lipid homeostasis in the mismatch groups suggests an increased lipid metabolism. While adipocytes are the main site of fatty acid and lipid metabolism, the liver is important in these processes, and changes in its transcriptome can cause a difference in intramuscular fatty acid composition [35]. As PROX1, HNF4α and FOXA1 were increased in the mismatch groups, the accumulation of lipids and triglycerides was relatively low. These groups demonstrated a need to increase lipid metabolism while the match groups did not. We also found additional genes important in lipid homeostasis (as mentioned in Armour et al. 2017 [34]) which were increased in the mismatch groups, such as integrator complex subunit 12 (INTS12) (FDR = 0.04) and CCAAT enhancer-binding protein delta (CEBPD) (FDR = 0.06). Horodyska et al. [32] studied pigs until slaughter, grouped them according to their feed efficiency, and compared the liver transcriptome by studying the biological processes relevant to the difference in feed efficiency. In the more feed-efficient pigs, they found an increased bile acid secretion, an increase in cell differentiation, a higher number of NK cells, an increased protein turnover (metabolism), and an immune response activation [32]. This appears to be linked to the decreased feed efficiency in the mismatch groups from up to 15 weeks of age [19]. These results here discussed suggest increased protein catabolism, which we hypothesize to have eventually led to lower feed efficiency, but also less lipid accumulation and less conversion of cholesterol to bile acids. All of these processes together may have contributed to worse feed efficiency. 

Serum Amyloid A2 (SAA2) was one of the most upregulated genes in the mismatch compared to the match groups. It is one of the four known SAA proteins, which are acute-phase proteins (APPs) upregulated at the sign of infection and often measured in serum [36,37]. In pigs, three isoforms of SAA circulate: SAA2, SAA3, and SAA4, with SAA2 being most prominent during an acute phase response [38]. In contrast to the gene expression results, ELISA data of non-specific SAA showed that the only tendency to increase in the serum was in the HL group, owing to many low and almost zero values. In a state of homeostasis, SAA isoforms circulate in low, negligible concentrations of 0.16 μg/mL for SAA2, 0.007 μg/mL for SAA3 and 0.032 μg/mL for SAA4 [38]. The upregulation of SAA2 genes without the protein synthesis in serum might mean that protein synthesis has not happened yet or the proteins have already been catabolized. The origin of stress could be the resilience to changes in diet [39]. There are indications that SAA plays multiple roles in infection, but since the serum values were close to homeostasis and because no related cytokines (IL-1, IL-6, and TNFα) are differentially expressed in the liver, the observed changes were not likely to be a direct response to an infection. If there were a response from feed stress or a mismatch in metabolic programming, then the SAA in serum could increase slightly. SAA correlates with obesity and insulin resistance in humans and responds quickly to high-fat diets and insulin response in mice [36,40]. The interaction between maternal and weaning diets might evoke a more sensitive response to inflammation stimuli or lead to higher levels of base APP. Since an immune response takes energy from growth and therefore performance, a mismatched diet might cause an allocation of energy from growth to APP production. As mentioned earlier, Horodyska et al. [32] have already demonstrated some evidence of this. Chen, et al. [41] demonstrated that a PUFA (polyunsaturated fatty acid)-induced SAA gene expression in hepatic tissue decreased PPARγ expression and consequently upregulated fatty acid oxidation and downregulated the expression of several genes involved in lipogenesis [41]. 

A shortcoming of this study was the use of both sexes at 3.5, but not at 4.5 weeks of age. Males and females have different metabolisms and the liver is highly sexually dimorphic which is largely caused by gonadal and consequently growth hormones [42,43]. More than 800 genes are known to be sexually dimorphic. This not only causes differences in steroid and hormonal metabolism but also in nuclear factors, receptors, signaling molecules, enzymatic and secretory pathways [42]. In more general terms, these genes are involved in sexual reproduction, lipid metabolism, and cardiovascular disease [44]. Therefore, it was important to statistically correct for sex at 3.5 weeks of age, however, no differences between treatments were found at this time point. FOXA1 is known to differ between sexes [43]. SAA (measured in serum) is also sexually dimorphic [45]. However, these genes were found as DEGs in our study at 4.5 weeks of age where we only have males investigated.

RYBP (Ring1 and YY1 Binding Protein) was another gene upregulated in the mismatch groups and we validated this gene because of its role in epigenetics. RYBP is a conserved epigenetic factor with both epigenetic silencer and activator mechanisms and has control over ubiquitin-binding activity, which is crucial for the developmental and functioning of epigenetic regulators, apoptosis, immune response, and other regulatory functions [46,47]. The presence of interaction effects in piglets demonstrates the possibility of an epigenetic mechanism involved due to diet during late gestation. The RYBP demonstrates a possible effect of fetal programming due to the late gestation diet, even though the effect is limited. It appears that this gene is of no importance to lipid metabolism or immune responses like the other validated genes, but is nonetheless supportive of our main conclusion. 

### 4.3. Main Effects of Dietary Treatment

Several genes and pathways appeared to differ depending on piglet diet, as indicated by piglet treatment as the main effect (Appendix A). In piglets receiving high protein diets, the upregulation in glucagon (KEGG:04922) and adipocytokine (KEGG:04920) appears to be driven by the genes G6PC1 (glucose-6-phosphatase catalytic subunit 1) (FDR = 0.10), an enzyme involved in gluconeogenesis, converting G6P to glucose, and CPT1A (carnitine palmitoyltransferase 1A) (FDR = 0.10) that oxidizes fatty acid to Acyl-CoA, shuttling fatty acids from the cytoplasm into the mitochondria [48]. Both G6PC1 and CPT1A are involved in increasing energy availability in the body. In these iso-energetic feeds, the difference is likely to come from different components in the diet that activate different biological mechanisms. 

From our H protein piglet diet, we observed an upregulation of the histidine ammonia-lyase activity (HAL) pathway (GO:0004397) that was only represented by one DEG from our data and is therefore not differentially expressed according to our criteria (data not shown). However, we wanted to discuss it because of its importance in protein metabolism. In our previous publication, we found a significant increase in serum urea for both sow and piglet H diets (HH = 23.91, HL = 13.84, LH = 20.24, LL = 8.84 mg/dL; SEM = 1.45; *P*_sow_ = 0.002 and *P*_piglet_ < 0.001 [19]). Pigs can convert accumulations of NH_3_ + H_2_O into the more stable urea molecule to transport it to the kidneys via blood through the HAL pathway [49]. The increased serum urea levels in piglets receiving a high piglet protein diet illustrate the breakdown of excess protein in these piglets [19], which is in agreement with the upregulation of HAL for piglets on the H diet (Appendix A).

There are three DEGs for the main effect of sows at 3.5 weeks and zero at 4.5 weeks, which shows that there is no general effect of maternal diet. Similar minor results were found in Schroyen et al. [50] where different sources of starch were provided in the sows’ diet throughout late gestation and lactation. Maternal diets contained either digestible starch or resistant starch and were fed to sows from day 88 until the end of lactation, which resulted in only a few differentially expressed genes in the liver and colon of piglets due to this maternal treatment. In addition, the study by Rooney et al. [51] provided four diets differing in energy density from day 108 of gestation and during lactation. They found differences in gene expression in the jejunum of sows up to 20 days of lactation but did not find any in neonatal piglets or piglets 7 days after weaning [51]. It is not uncommon that effects disappear after only a short treatment period, or that they only appear after a given time. During late gestation, the fetus has fully developed and nutrients and energy are dedicated to the growth and mammary development. Therefore, only a limited effect of late gestation treatment is to be expected. The presence of interaction effects but the absence of major main treatment effects in piglets demonstrates the possibility of an epigenetic mechanism involved due to diet during late gestation, albeit possibly with only a temporary effect.

## 5. Conclusions

Gene expression differences were found, with differences between match and mismatch groups, demonstrating that the sow diet could influence how piglets metabolize their diet. Several upregulated genes were linked to either protein-related catabolic pathways, inflammation, or lipid metabolism genes, but were mostly general and indirective. The analyses found differentially expressed genes that could explain why we previously found a decreased feed efficiency in mismatch piglet feeding groups. These DEGs for interaction effect suggest that the maternal protein late gestation diet potentially has an impact on the metabolism of piglets and should be considered in feeding strategies.

## Figures and Tables

**Figure 1 vetsci-08-00233-f001:**
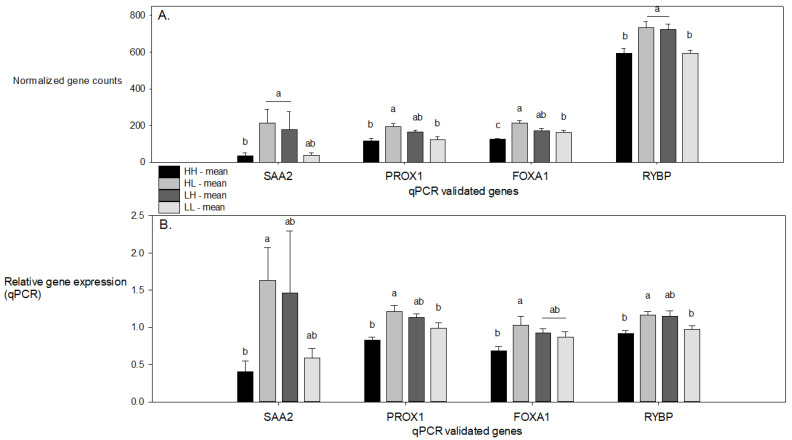
(**A**) Normalized gene counts of differential expressed genes (DEGs) in the RNA-seq counts for the four genes validated with qPCR with a standard error to the mean (SEM) indicated. Results are shown for each of the four treatment groups HH = higher (late gestation sow diet) − higher (nursery diet piglet) CP protein, HL = higher − lower CP protein, LH = lower − higher CP protein, and LL = lower − lower CP protein. (**B**) Relative DEGs for qPCR results. The significant differences between treatments (*p* < 0.05) are denoted by a and b, were ab was not significantly different from either a or b.

**Table 1 vetsci-08-00233-t001:** Differential expressed genes at 4.5 weeks of age between the match (LL and HH) and mismatch groups (HL and LH) describing the interaction effect between piglets during the nursery phase and maternal diet during late gestation. Average expression is the mean of normalized counts of all samples.

Ensembl Gene ID	Gene Symbol	Average Expression	Description
ENSSSCG00000015480	*PRRC2C*	801	Downreg. in mismatch
ENSSSCG00000008735	*BOD1L1*	330	Downreg. in mismatch
ENSSSCG00000022828	*PNN*	366	Downreg. in mismatch
ENSSSCG00000001002	*PRPF4B*	588	Downreg. in mismatch
ENSSSCG00000035080	*RPL23A*	4872	Downreg. in mismatch
ENSSSCG00000021363	*CHD3*	178	Downreg. in mismatch
ENSSSCG00000016846	*WDR70*	71	Downreg. in mismatch
ENSSSCG00000005373	*NANS*	474	Downreg. in mismatch
ENSSSCG00000033591	*CHD9*	198	Downreg. in mismatch
ENSSSCG00000003791	*SRSF11*	835	Downreg. in mismatch
ENSSSCG00000040118	*PPDPF*	1845	Downreg. in mismatch
ENSSSCG00000011284	*NKTR*	378	Downreg. in mismatch
ENSSSCG00000001930	*PKM*	157	Downreg. in mismatch
ENSSSCG00000021138	*CEP250*	64	Downreg. in mismatch
ENSSSCG00000001929	*-*	33	Downreg. in mismatch
ENSSSCG00000011278	*TRAK1*	89	Downreg. in mismatch
ENSSSCG00000038682	*TAF1D*	315	Downreg. in mismatch
ENSSSCG00000010926	*SYT2*	143	Downreg. in mismatch
ENSSSCG00000015035	*C11orf52*	96	Downreg. in mismatch
ENSSSCG00000025272	*HEMK1*	80	Downreg. in mismatch
ENSSSCG00000016123	*PARD3B*	155	Downreg. in mismatch
ENSSSCG00000000695	*IFFO1*	45	Downreg. in mismatch
ENSSSCG00000033636	*AFMID*	267	Downreg. in mismatch
ENSSSCG00000013145	*DTX4*	27	Downreg. in mismatch
ENSSSCG00000031259	*snoU2-30*	387	Downreg. in mismatch
Ensembl gene ID	Gene Symbol	Average expression	Description
ENSSSCG00000036684	*TTPAL*	41	Upreg. in mismatch
ENSSSCG00000026606	*PAIP2*	211	Upreg. in mismatch
ENSSSCG00000009772	*TMED2*	524	Upreg. in mismatch
ENSSSCG00000037634	*FOXA1*	168	Upreg. in mismatch
ENSSSCG00000013370	*SAA2*	116	Upreg. in mismatch
ENSSSCG00000004414	*CD164*	462	Upreg. in mismatch
ENSSSCG00000035774	*ERRFI1*	371	Upreg. in mismatch
ENSSSCG00000002867	*CEBPG*	67	Upreg. in mismatch
ENSSSCG00000002351	*PTGR2*	245	Upreg. in mismatch
ENSSSCG00000006289	*F5*	639	Upreg. in mismatch
ENSSSCG00000024019	*GTF2H5*	91	Upreg. in mismatch
ENSSSCG00000011367	*ARIH2*	63	Upreg. in mismatch
ENSSSCG00000032580	*MGST1*	882	Upreg. in mismatch
ENSSSCG00000030617	*INTS12*	9	Upreg. in mismatch
ENSSSCG00000005994	*SNTB1*	128	Upreg. in mismatch
ENSSSCG00000025053	*RYBP*	661	Upreg. in mismatch
ENSSSCG00000013060	*SCGB1A1*	14	Upreg. in mismatch
ENSSSCG00000027157	*SLC40A1*	1103	Upreg. in mismatch
ENSSSCG00000015584	*PROX1*	148	Upreg. in mismatch
ENSSSCG00000014948	*C11orf54*	119	Upreg. in mismatch
ENSSSCG00000011695	*AGTR1*	113	Upreg. in mismatch
ENSSSCG00000004209	*PTPRK*	142	Upreg. in mismatch

**Table 2 vetsci-08-00233-t002:** Pathway analyses at 4.5 weeks of age describing the interaction effect between piglet diet during the nursery phase and maternal diet during late gestation at an FDR < 0.10. Up- and down-regulated genes were analyzed separately (a) and together (b). (a): pathway analysis for either up- or downregulated genes in the mismatched group. (b): pathway analysis for both up- and downregulated genes together.

**(a)**					
**GO ID**	**Source**	**Description**	**FDR**	**Genes**	**Description**
KEGG:03010	KEGG	Ribosome	0.000	8	Downreg. in mismatch
KEGG:05171	KEGG	Coronavirus disease-COVID-19	0.020	8	Downreg. in mismatch
GO:0006511	GO:BP	Ubiquitin-dependent protein catabolic process	0.000	14	Upreg. in mismatch
GO:0019941	GO:BP	Modification-dependent protein catabolic process	0.000	14	Upreg. in mismatch
GO:0030970	GO:BP	Retrograde protein transport, ER to cytosol	0.000	5	Upreg. in mismatch
GO:0043632	GO:BP	Modification-dependent macromolecule catabolic process	0.000	14	Upreg. in mismatch
GO:1903513	GO:BP	Endoplasmic reticulum to cytosol transport	0.000	5	Upreg. in mismatch
GO:0032527	GO:BP	Protein exit from the endoplasmic reticulum	0.010	5	Upreg. in mismatch
GO:0044281	GO:BP	Small-molecule metabolic process	0.010	23	Upreg. in mismatch
GO:0010498	GO:BP	Proteasomal protein catabolic process	0.020	11	Upreg. in mismatch
GO:0043161	GO:BP	Proteasome-mediated ubiquitin-dependent protein catabolic process	0.020	10	Upreg. in mismatch
GO:0051603	GO:BP	Proteolysis involved in cellular protein catabolic process	0.020	14	Upreg. in mismatch
GO:0030323	GO:BP	Respiratory tube development	0.030	8	Upreg. in mismatch
GO:0044248	GO:BP	Cellular catabolic process	0.030	24	Upreg. in mismatch
GO:1901575	GO:BP	Organic substance catabolic process	0.030	24	Upreg. in mismatch
KEGG:04141	KEGG	Protein processing in endoplasmic reticulum	0.030	7	Upreg. in mismatch
**(b)**					
**GO ID**	**Source**	**Description**	**FDR**	**Genes**	**Description**
GO:0008152	GO:BP	Metabolic process	0.0000648	166	Downregulated
GO:0044237	GO:BP	Cellular metabolic process	0.0003	152	Downregulated
GO:0044238	GO:BP	Primary metabolic process	0.0003	152	Downregulated
GO:0071704	GO:BP	Organic substance metabolic process	0.0003	159	Downregulated
GO:1901564	GO:BP	Organonitrogen compound metabolic process	0.023	100	Downregulated
GO:0035904	GO:BP	Aorta development	0.043	6	Downregulated
GO:0006807	GO:BP	Nitrogen compound metabolic process	0.047	138	Downregulated
KEGG:05171	KEGG	Coronavirus disease-COVID-19	0.013	12	Downregulated

**Table 3 vetsci-08-00233-t003:** Mean values for the SAA ELISA serum concentrations at 3.5 weeks of age of the piglets with only a sow late gestation treatment at weaning.

Sow Feed:	Higher Protein (H) ^1^		Lower Protein (L) ^1^		*P* _sow_
	Mean	Se	Mean	Se	
SAA ELISA 3.5 weeks (μg/mL)	37.48	32.47	5.09	2.64	0.49

^1^ At 3.5 weeks of age, *n* = 12 per treatment.

**Table 4 vetsci-08-00233-t004:** Mean values for the SAA ELISA serum concentrations at 4.5 weeks of age of the piglets with both sow and piglet treatment at 4.5 weeks of age.

Piglet Feed:	Higher Protein ^2^ (HH)		Lower Protein ^2^ (HL)		Higher Protein ^2^ (LH)		Lower Protein ^2^ (LL)		*P* _sow_	*P* _piglet_	*P* _s*p_ ^1^
	Mean	Se	Mean	Se	Mean	Se	Mean	Se			
SAA ELISA 4.5 weeks (μg/mL)	1.71	1.32	11.17	9.03	5.76	4.93	0.68	0.20	0.11	0.62	0.07

^1^ P_s*p_ denotes the *p*-value for the interaction between the piglet and sow diet. ^2^ At 4.5 weeks of age, *n* = 8 for HH, LH, and LL and *n* = 7 for HL.

## Data Availability

All data is available upon request.

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
