# Peer review of "The Impact of Maternal and Piglet Low Protein Diet and Their Interaction on the Porcine Liver Transcriptome around the Time of Weaning"

_vetsci, 2021, doi:10.3390/vetsci8100233_

Round 1

Reviewer 1 Report

1.It is suggested that phenotypic indicators in the relevant blood be added to support the speculations of 4.3 in the discussion
2.why choose FDR<0.10 as pathway analyses used genes ?
3.Where are the primers designed in this paper
4.Of the four genes selected, the RYBP gene did not seem to advance the conclusion of the experiment any more than the other three

Author Response

  1. It is suggested that phenotypic indicators in the relevant blood be added to support the speculations of 4.3 in the discussion

Previously we found a significant increase in serum urea for both sow and piglet H diets (HH=23.91, HL=13.84, LH=20.24, LL=8.84 mg/dL; SEM=1.45; Psow=0.002 and Ppiglet<0.001 [19]). We have added this specifically in this article as well, §4.3 line 387-388. If these are not the phenotypical indicators referred to, please let us know.

  1. Why choose FDR<0.10 as pathway analyses used genes?

We did that as to get more genes to do a proper, more meaningful, pathway analysis. We considered a gene DE of FDR<0.05, but when we were that strict when trying to look for pathways, we would have been too severe as in this experiment changes seem very subtle. By broadening pathway analyses to FDR<0.10 a clearer view of potential mechanisms might show up, but we clearly indicate that we lower our threshold for doing such analyses. However, in Table 1 for example, we only show the DE genes, with an FDR<0.05. Moreover, in Table 2, where we show the pathway analyses (which we now reduced to KEGG, GO: Biological Processes and GO:Molecular Function), we only take those pathways with a minimum of 5 genes present with a FDR<0.10 for differential expression. The FDR for the pathway analyses given by g:profiler2 is also always shown in this table, and that is always <0.05.

  1. Where are the primers designed in this paper?

We stated this in the material and methods (§2.4, line 160-161) and in Suppl. Table 1. We designed most of our primers with Primer3 and we used one from literature.

  1. Of the four genes selected, the RYBP gene did not seem to advance the conclusion of the experiment any more than the other three

We thought initially that this gene would be interesting because of its role in epigenetics. We agree that this gene is unsupportive of our conclusion and is not involved in lipid metabolism and immune response mechanisms. Nevertheless, it was interesting to see that it was upregulated in our mismatch groups in the RNA-seq, and significantly upregulated in the HL group when compared to the LL and HH group using qPCR. We elaborated a little on its epigenetic role, which is interesting in our fetal programming experiment (§4.2, line 366-370).

Reviewer 2 Report

The manuscript titled “The impact of maternal and piglet low protein diet and their interaction on the porcine liver transcriptome around the time of weaning” submitted by Kroeske et al., is an excellent manuscript that I recommend to be accepted with minimal revisions.

The topic of the research is highly relevant to the industry and pig welfare, experimental design is solid and well presented, methods are very deeply explained and justified. The analysis is done at a high level using the best methods in the area, the transcriptome data was further validated by PCR and supported with immunoassay.  

Results are well presented, figures well made and well-chosen,  and discussion goes very deep into interpretation without overstating the findings.

Since I felt I have to find something to correct, to show I seriously reviewed this paper, I have a few comments for the methods section:  how were the sequences filtered and processed and how many seqs per sample? What was qual score cutoff etc.  Were sequences 75nt after 3 and 5’ cleanup? Was there a statistical difference in seqs number between the groups?

Finally, I want to wish the authors much success in their future work. This paper is a fine contribution to pig health and nutrition science.

Author Response

The manuscript titled “The impact of maternal and piglet low protein diet and their interaction on the porcine liver transcriptome around the time of weaning” submitted by Kroeske et al., is an excellent manuscript that I recommend to be accepted with minimal revisions.

The topic of the research is highly relevant to the industry and pig welfare, experimental design is solid and well presented, methods are very deeply explained and justified. The analysis is done at a high level using the best methods in the area, the transcriptome data was further validated by PCR and supported with immunoassay.  Results are well presented, figures well-made and well-chosen, and discussion goes very deep into interpretation without overstating the findings.

Since I felt I have to find something to correct, to show I seriously reviewed this paper, I have a few comments for the methods section:  how were the sequences filtered and processed and how many seqs per sample? What was qual score cutoff etc.  Were sequences 75nt after 3 and 5’ cleanup? Was there a statistical difference in seqs number between the groups?

  • We rewrote part of §2.2 to: The sequencing yielded 6,85E+06 ± 1,72E+06 (mean ± standard deviation) raw reads per sample. The mean Phred score across each base position in the reads was >28 for all samples (Supplementary Figure 1). The raw reads were processed by trimming the adapter sequences and polyA tail sequences using Trim Galore{The Babraham Bioinformatics group,  #1515}. The length distribution of the remaining reads can be found in Supplementary Figure 2. The trimmed reads were filtered to remove rRNA reads using Bowtie2 {Langmead, 2019 #1514} and a custom-built reference database with all pig ribosomal RNA sequences found on SILVA {Quast, 2012 #1509;SILVA,  #1510}. The 6,67E+06 ± 1,67E+06 remaining reads were mapped against the Sus scrofa (v11.1) reference genome {Warr, 2020 #1516} from the Ensembl genome database using the STAR mapper {Dobin, 2015 #1517}. On average, 78,51% of the reads mapped uniquely to the reference genome, 87,12% of the reads mapped uniquely or to a limited number of loci in the reference genome.

Finally, I want to wish the authors much success in their future work. This paper is a fine contribution to pig health and nutrition science.

  • Thank you very much

Reviewer 3 Report

The manuscript deals with the feeding of low and high protein diets to gestating sows and their offspring.  It is intended to complement an earlier work, which provides zootechnical parameters and digestibility data, investigating the hypothesised shift in offspring liver metabolism at the transcriptome level. There are a number of studies with a very similar focus, but especially this topic of metabolic programming needs repeated observations to back up the findings. However, even if the study of gene expression is not a major challenge nowadays, the main difficulty is the adequate analysis and interpretation of the data, which has clear weaknesses in the present work.

The pathway analysis is flawed and needs to be revised. First, it needs to be decided which database is included. Given the objective of the study, KEGG and GO:MF are likely to be the most appropriate, while at least GO:CC does not make sense. Secondly, why are up-regulated and down-regulated genes analysed separately? Both activation and inhibition of a signalling pathway are conceivable. Third, adjusting for multiple testing should also be considered for pathway analysis. Fourth, where is the enrichment in signalling pathways represented by a small number of DEGs (e.g. ≤ 3)? These cases should already be excluded in the M&M description.

Taking into account these criteria and the resulting findings, the discussion needs substantial revision. Moreover, it must be taken into account that the enriched processes are essentially intracellular pathways. Thus, the protein catabolism mentioned includes intracellular degradation via proteasomes and has nothing to do with the utilisation of dietary protein. In addition, it needs to be considered that lipogenesis does not take place in the liver of pigs.

The current conclusion is not clearly supported by the data. No "clear differences" have been demonstrated at transcriptional level and the argumentation for the molecular processes influencing the differences in feed efficiency is very vague.

More specific comments:

Why did authors choose 3.5 weeks and 4.5 weeks of age for gene expression analysis. Is it assumed that 1 week of postnatal diet considerably changes the liver metabolism?

Why only males were investigated at 4.5 weeks of age?

The sow might have a strong effect on the offspring. It is unclear what is the maternal origin of the select piglets. Why sow was not considered in the statistical model?

Line 77: Relation to CP is missing.

Line 107: Information about sequencing should be provided after library preparation.

Line 111: Unclear if libraries were prepared for each sample or pooled.

Line 117: More details for read processing and mapping need to be provided.

RNA-seq differential expression analysis: Important information such as the statistical test used, the average number of reads per sample and whether the raw data is of good quality is missing.

Line 121: Change “compute” to “identify”

Formula (2): Change to “Dietsow”. It is unclear how match vs. mismatch analyses were performed as the interaction of Dietsow and Dietpiglet will result in 4 groups.

Line 131: If filtering is done to remove observations with very low abundance, this must be done before the DEG analysis based on the read counts. As it is, it is not appropriate.

Line 136: Details on the usage of gprofiler 2 are missing (e.g. version, type of input gene IDs, data bases, settings).

Line 160: Normalization of RNA-seq counts needs to be specified.

Line 165: It is unclear whether the dilutions were carried out consecutively or whether other criteria were used for dilution.

Line 172: No statistical analysis of the SAA values are mentioned.

E.g. line 188: Links to some references are missing.

Line 207: Does this refers to raw counts? What is the error indicator in Figure 1?

Table 1: What is represented by the “Average expression” column?

Author Response

Dear reviewer, 

Our sincerest gratitude is given to the reviewers for their constructive feedback. We have addressed all the comments to improve our manuscript. We hope that the changes made will be received as favourable enabling the acceptance of our manuscript. In what follows we give a rebuttal with the answer to every question in green, and we used track changes to change the manuscript in order to make it easy for the reviewers to see the differences. Line numbering in the rebuttal is based on ‘simple markup’ settings in the review pane of the manuscript.

Kind regards,

the authors

----

  • The manuscript deals with the feeding of low and high protein diets to gestating sows and their offspring.  It is intended to complement an earlier work, which provides zootechnical parameters and digestibility data, investigating the hypothesised shift in offspring liver metabolism at the transcriptome level. There are a number of studies with a very similar focus, but especially this topic of metabolic programming needs repeated observations to back up the findings. However, even if the study of gene expression is not a major challenge nowadays, the main difficulty is the adequate analysis and interpretation of the data, which has clear weaknesses in the present work.
  1. The pathway analysis is flawed and needs to be revised. First, it needs to be decided which database is included. Given the objective of the study, KEGG and GO:MF are likely to be the most appropriate, while at least GO:CC does not make sense.

Indeed GO:CC is never very informative, so we can leave that GO source out. We kept KEGG, GO:BP and GO:MF when present. Table 2 is therefore shorter, more focusing on meaningful GO terms. We omitted Suppl Table 4 as not interesting pathways came out of that anymore, only the HAL pathway that we still discuss in our discussion. The HAL gene was indeed also one of our DEG and we refer to that one instead of the pathway.

  1. Secondly, why are up-regulated and down-regulated genes analysed separately? Both activation and inhibition of a signalling pathway are conceivable.

Thank you for this remark. We have changed the analysis and added the results as requested to Table 2 (line 245). We now have Table 2a with GO terms from individual analyses, and Table 2b with GO terms when taking all DEG together.

  1. Third, adjusting for multiple testing should also be considered for pathway analysis.

The FDR for the pathway analyses given by g:profiler2 is also always shown in these tables, and that is always <0.05. We made sure this is clear in the tables now.

  1. Fourth, where is the enrichment in signalling pathways represented by a small number of DEGs (e.g. ≤ 3)? These cases should already be excluded in the M&M description.

Thank you for this remark. We added this to the M&M and removed all pathways with less than 5 differentially expressed genes representing them. This made us not show Suppl Table 4 (pathways for main effects) in the final manuscript.

  1. Taking into account these criteria and the resulting findings, the discussion needs substantial revision. Moreover, it must be taken into account that the enriched processes are essentially intracellular pathways. Thus, the protein catabolism mentioned includes intracellular degradation via proteasomes and has nothing to do with the utilisation of dietary protein. In addition, it needs to be considered that lipogenesis does not take place in the liver of pigs.

Thank you for this critical remark. We have adjusted §4.2 according to the directions provided. Indeed, lipogenesis occurs mainly in the adipocytes. However, the liver is also involved in fatty acid metabolism. We added this in (§4.2, line 312-319).

  1. The current conclusion is not clearly supported by the data. No "clear differences" have been demonstrated at transcriptional level and the argumentation for the molecular processes influencing the differences in feed efficiency is very vague.

Thank you for this critical remark. We did find some interesting genes in meaningful pathways, however, we nuanced our conclusions. We say that: ”These DEGs for interaction effect suggest that the maternal protein late gestation diet potentially has an impact on the metabolism of piglet and should be considered in feeding strategies.” (5. Conclusion, line 411-418).

More specific comments:

  • Why did authors choose 3.5 weeks and 4.5 weeks of age for gene expression analysis. Is it assumed that 1 week of postnatal diet considerably changes the liver metabolism?

We chose two time points close around weaning as we wanted to see if there were some direct changes before and after this critical time point. However, indeed, it would be interesting to look at other time points, longer after weaning, and to see if there would be a difference in metabolic changes, that might not have been present at 4.5 weeks yet.

  • Why only males were investigated at 4.5 weeks of age?

The short answer is that we tried, whenever possible, to limit the sex factor in our analyses. The long answer is: our experiment consisted of three parts: 1. The performance- (until 24 weeks of age), 2. The transcriptomics-, and 3. The metagenomics part. It was important for the performance part of our experiment to limit the gender factors and for the microbiome part to keep litters together. For the long time performance we chose to follow up the gilts and we only had six piglets in each pen after weaning of which 3 were gilts. This left us with the castrated males for sampling for our transcriptomic analyses at 4.5 weeks. We had to sacrifice both sexes at 3.5 weeks of age to get to sufficient animals at this age.

  • The sow might have a strong effect on the offspring. It is unclear what is the maternal origin of the select piglets. Why sow was not considered in the statistical model?

There were too many sows to take this into account in our statistical model since we selected a maximum of one piglet per pen per sampling (and we kept the litters together, so each pen is a different sow). However, we added boar effect because we only used two boars in this experiment.

  • Line 75: Relation to CP is missing.

We have added this.

  • Line 116: Information about sequencing should be provided after library preparation.

Thank you for your remark. We added this in §2.2. It was as also suggested by reviewer 2.

  • Line 111: Unclear if libraries were prepared for each sample or pooled.

Samples were not pooled. We now also clarified this in §2.2.

  • Line 117: More details for read processing and mapping need to be provided.

This was added as well in §2.2.

  • RNA-seq differential expression analysis: Important information such as the statistical test used, the average number of reads per sample and whether the raw data is of good quality is missing.

This was added to §2.2 as well.

  • Line 129: Change “compute” to “identify”

We changed this.

  • Formula (2): Change to “Dietsow”. It is unclear how match vs. mismatch analyses were performed as the interaction of Dietsow and Dietpiglet will result in 4 groups.

Since the result of Formula 2 would result in four treatment groups, the list of genes was exported, visualized and the results were categorized as ‘match’ or ‘mismatch’. We added this in §2.3.

  • Line 130: If filtering is done to remove observations with very low abundance, this must be done before the DEG analysis based on the read counts. As it is, it is not appropriate.

We mentioned this before in line 130, but we have added ‘before the DEG analysis’ to clarify this.

  • Line 136: Details on the usage of gprofiler 2 are missing (e.g. version, type of input gene IDs, data bases, settings).

Line 148-150: g:Profiler2 (version 0.2.1). Ensemble identifiers were directly imported from DESeq2 (DESeq large dataframe), using the g:Profiler database. the -log10(p-values) were capped, phenotypes were set to +1/-1.  We added this to clarify.

  • Line 160: Normalization of RNA-seq counts needs to be specified.

Standard normalization with DESeq2 was applied. The data was normalized by the DEseq2 function ‘DESeq’ with a normalization factor sij, i.e., μij = sijqij. We added this to the paper in line 130.

  • Line 165: It is unclear whether the dilutions were carried out consecutively or whether other criteria were used for dilution.

We added on line 194: Dilutions were determined based on signal and repeated if no or inconsistent signal.

  • Line 172: No statistical analysis of the SAA values are mentioned.

We added this to line 186-189 (§2.5): “Results were statistically analysed using numSummary (RcmdrMisc, R), and mixed models (lme4, R) with the interaction effect of piglet and sow diet (multiplied within the model) as fixed effects and Batch as a random effect with a significance threshold of P<0.05.”

  • E.g. line 188: Links to some references are missing.

Thank you for the comment. This is fixed now.

  • Line 207: Does this refers to raw counts? What is the error indicator in Figure 1?

We added standard error to the mean (SEM) as error indicator.

  • Table 1: What is represented by the “Average expression” column?

It is the mean of normalized counts of all samples. We added ‘Average expression is the mean of normalized counts of all samples’ to the header of the Table.

Round 2

Reviewer 3 Report

The authors have revised or clarified several points that were noted in the previous revision. The pathway analyses are now acceptable. However, the DEG analysis still does not appear to have been carried out according to standard procedures.

Line 141: The comparison of the 4 treatment groups results in 6 contrasts. Each contrast has a fold change and p and q values. Out of these 6 six contrasts, LL and HH as well as LH and HL were summarized to “match” and “mismatch”. How is it then possible to determine a single log2FC, p-value and FDR for "match" vs. "mismatch" as shown in Table 1? Regardless of any explanation, the contrast of interest "match" versus "mismatch" must be defined in the DESeq model and not assembled afterwards from individual analyses.

Line 92: Since the manuscript is about foetal programming, it is absolutely necessary to mention that all 12 animals belonging to one group (L) and (H) originate from different sows. The same applies to the 4.5-week-old animals.

Line 143: Sentence needs to be rephrased as “expression level” should be “count”.

Line 149: Ensembl ID originate from STAR mapping, not from DESeq; the gene IDs might be revealed via g:Profiler2.

Line 150: This sentence is unclear: “The -log10 (p-values) were capped and phenotypes were set to +1/-1.”

Line 153: Change “genes” to “DEG”.

Line 236: Figure 1A shows absolute and not relative gene expression data. In general, for calculation purposes (e.g. correlation) count data transformation should be performed according to the DESeq2 vignette (e.g. pseudo counts, rlogs, vst).

The intention to limit the effect of gender in the study has obviously failed, therefore the differences in the gender composition between 3.5 weeks and 4.5 weeks and the possible consequences for the outcome should be addressed in the discussion.

Author Response

Dear reviewer, find the cover letter and the rebuttal also in the attachment. Kind regards, the authors. 

---- 

Dear reviewer,

Our sincerest gratitude is given to you for your constructive feedback. We have addressed all the comments to improve our manuscript. We hope that the changes made will be received as favourable enabling the acceptance of our manuscript. In what follows we give a rebuttal with the answer to every question in blue, and we used 'track changes' to change the manuscript in order to make it easy for the reviewers to see the differences. The editor marked all previous revisions in yellow. Line numbering in the rebuttal is based on ‘simple markup’ settings in the review pane of the manuscript.

Kind regards,

The authors.

-----

The authors have revised or clarified several points that were noted in the previous revision. The pathway analyses are now acceptable. However, the DEG analysis still does not appear to have been carried out according to standard procedures.

  • Line 141: The comparison of the 4 treatment groups results in 6 contrasts. Each contrast has a fold change and p and q values. Out of these 6 six contrasts, LL and HH as well as LH and HL were summarized to “match” and “mismatch”. How is it then possible to determine a single log2FC, p-value and FDR for "match" vs. "mismatch" as shown in Table 1? Regardless of any explanation, the contrast of interest "match" versus "mismatch" must be defined in the DESeq model and not assembled afterwards from individual analyses.

In the model DESeq2 model design 4.5 weeks of age = ~Batch+Boar+Dietsow+Dietpiglet+Dietsow:Dietpiglet the output reflects those genes that show a clear (significant to a level of FDR <0.05) interaction effect between the diet of the sow and the diet of the piglet because the colon mark indicates the interacting variables within the formula . Indeed, the reviewer is right that it makes not much sense to have a log2FC (and SE log2FC) shown for this interaction effect, as it is not clear what is higher or lower expressed, and what directs us to have a significant interaction effect. We therefore adjusted Table 1 and left out the log2FC and SElog 2FC.

However, we do believe that it was useful to go and look at this relatively short list of genes and plot counts individually for each gene listed. We thereafter could classify these genes as clearly either down- or upregulated in the LH and HL versus LL and HH. We do believe that this information is interesting for the readers to know.

  • Line 92: Since the manuscript is about foetal programming, it is absolutely necessary to mention that all 12 animals belonging to one group (L) and (H) originate from different sows. The same applies to the 4.5-week-old animals.

Thank you for this critical remark. We added to line 91-92: “all 24 originate from different sows,” and to line 93 “all from different sows”.

  • Line 143: Sentence needs to be rephrased as “expression level” should be “count”.

We changed this to ‘counts’, the new line number is 144.

  • Line 149: Ensembl ID originate from STAR mapping, not from DESeq; the gene IDs might be revealed via g:Profiler2.

We changed this to ‘STAR mapping’, the new line number is 150.

  • Line 150: This sentence is unclear: “The -log10 (p-values) were capped and phenotypes were set to +1/-1.”

We agree. We changed the sentences and moved it to line 150: “DEGs from the interaction effect had an FDR<0.05. For the pathway analyses, genes were used with an FDR<0.10.”

  • Line 153: Change “genes” to “DEG”.

We changed this to ‘DEGs’, the new line number is 153.

  • Line 236: Figure 1A shows absolute and not relative gene expression data. In general, for calculation purposes (e.g. correlation) count data transformation should be performed according to the DESeq2 vignette (e.g. pseudo counts, rlogs, vst).

Relative was changed to normalized counts, new line: 233. Just to clarify, we also used normalized counts for downstream calculations

  • The intention to limit the effect of gender in the study has obviously failed, therefore the differences in the gender composition between 3.5 weeks and 4.5 weeks and the possible consequences for the outcome should be addressed in the discussion.

We added this alinea to the discussion (lines 358-369): A shortcoming of this study was the use of both sexes at 3.5, but not at 4.5 weeks of age. Males and females have different metabolisms and the liver is highly sexual dimorphic which is largely caused by gonadal and consequently growth hormones {Brie, 2019 #1521}{Yang, 2006 #1520}. More than 800 genes are known to be sexually dimorphic. This not only causes differences in steroid and hormonal metabolism, but also in nuclear factors, receptors, signalling molecules, enzymatic and secretory pathways {Brie, 2019 #1521}. In more general terms, these genes are involved in sexual reproduction, lipid metabolism and cardiovascular disease {Zhang, 2011 #1523}. Therefor it was important to statistical correct for sex at 3.5 weeks of age, however, no differences between treatments were found at this time point.  FOXA1 is known to differ between sexes {Yang, 2006 #1520}. Also SAA (measured in serum) is sexually dimorphic {Christoffersen, 2015 #727}. However, these genes were found as a DEGs in our study at 4.5 weeks of age where we only have males investigated.

Round 3

Reviewer 3 Report

Table 1 still contains p-values and FDR, for which it is unclear where they come from given the 6 possible contrasts. Consequently, either delete these columns and report only the genes that are differentially abundant in the comparison between match and mismatch, or clearly describe the methods and the origin of these p-values (e.g., statistical tests in DESeq with Wald test or LRT). In general, it is appreciated that authors accept comments and make efforts to improve the manuscript. However, it seems that the authors are somewhat reluctant to model match vs mismatch groups appropriately.

Author Response

Dear reviewer,

Thank you for your critical remark we adjusted the table as requested.

Kind regards, the authors.